# Strategies for Treating Traumatic Neuromas with Tissue-Engineered Materials

**DOI:** 10.3390/biom14040484

**Published:** 2024-04-16

**Authors:** Teng Wan, Qi-Cheng Li, Ming-Yu Qin, Yi-Lin Wang, Feng-Shi Zhang, Xiao-Meng Zhang, Yi-Chong Zhang, Pei-Xun Zhang

**Affiliations:** 1Department of Orthopedics and Trauma, Peking University People’s Hospital, Beijing 100044, China; 2111110396@bjmu.edu.cn (T.W.);; 2Key Laboratory of Trauma and Neural Regeneration, Peking University, Beijing 100044, China; 3National Centre for Trauma Medicine, Beijing 100044, China; 4Beijing Laboratory of Trauma and Nerve Regeneration, Peking University, Beijing 100044, China; 5Suzhou Medical College, Soochow University, Suzhou 215026, China; 6Peking University People’s Hospital Qingdao Hospital, Qingdao 266000, China

**Keywords:** tissue-engineered materials, neuroma, peripheral nerve injury, biomaterials, cell therapy, growth factors

## Abstract

Neuroma, a pathological response to peripheral nerve injury, refers to the abnormal growth of nerve tissue characterized by disorganized axonal proliferation. Commonly occurring after nerve injuries, surgeries, or amputations, this condition leads to the formation of painful nodular structures. Traditional treatment options include surgical excision and pharmacological management, aiming to alleviate symptoms. However, these approaches often offer temporary relief without addressing the underlying regenerative challenges, necessitating the exploration of advanced strategies such as tissue-engineered materials for more comprehensive and effective solutions. In this study, we discussed the etiology, molecular mechanisms, and histological morphology of traumatic neuromas after peripheral nerve injury. Subsequently, we summarized and analyzed current nonsurgical and surgical treatment options, along with their advantages and disadvantages. Additionally, we emphasized recent advancements in treating traumatic neuromas with tissue-engineered material strategies. By integrating biomaterials, growth factors, cell-based approaches, and electrical stimulation, tissue engineering offers a comprehensive solution surpassing mere symptomatic relief, striving for the structural and functional restoration of damaged nerves. In conclusion, the utilization of tissue-engineered materials has the potential to significantly reduce the risk of neuroma recurrence after surgical treatment.

## 1. Introduction

Neuroma is a benign, non-neoplastic proliferation of nerve fibers following injury or amputation. Its formation can be described as “a non-neoplastic response to nerve injury, resulting in reactive hyperplasia” or “the failure of the injured nerve to attempt regeneration” [1]. When a peripheral nerve is damaged, if the severed nerve ends are too far apart or there is a scar or other tissue barrier between them, or if the distal end is lost due to amputation, the regenerated axons fail to reach the distal end and intertwine with the proliferation of connective tissue, forming a coiled mass known as traumatic neuroma [2]. The prominent symptom is the existence of typical percussion pain and tenderness in the region of pain hypersensitivity (dysesthesia) or neuroma, which can impact the patient’s work and daily life, causing significant distress to the patient [3].

Current treatment options for painful neuromas include surgical interventions such as neuroma excision or nonsurgical approaches like pharmacological management, physical therapy, and neuromodulation techniques to alleviate symptoms and improve quality of life [4]. Nonsurgical interventions provide a holistic approach to managing pain in nerves related to traumatic neuromas [5]. Physical therapy is fundamental, concentrating on exercises to improve nerve gliding and mobility. Pharmacological options encompass analgesics, anti-inflammatory drugs, and neuropathic pain medications to relieve symptoms [6]. Furthermore, nerve blocks with local anesthetics or corticosteroids can offer temporary relief. Evidence-based therapies encompass peripheral nerve stimulation, spinal cord stimulation, and dorsal root ganglion stimulation [7]. If conservative measures are inadequate, surgical intervention becomes a crucial aspect of managing traumatic neuromas [8]. Neurectomy, the surgical removal of the neuroma, is a common technique aimed at eliminating the source of pain. However, this approach requires meticulous surgical skills to prevent damage to adjacent structures and the potential recurrence of neuromas [9]. Autograft and allograft techniques are currently regarded as the “gold standard” for treating repairable peripheral nerve defects where a tension-free, end-to-end anastomosis is not feasible. However, this approach is limited by a shortage of donors, potential failure of donor site function, and the risk of secondary injury [10]. Capping the injured peripheral nerve stump in the adjacent muscle and bone marrow cavity or adding a “lid” with autologous blood vessels to the nerve stump have also shown clinical efficacy [11]. However, it is still limited by a high recurrence rate, anatomical location, potential irritation of the injured nerve stump due to traction, issues with the immune microenvironment, and unfavorable three-dimensional physical topography for nerve repair [8].

Tissue-engineered materials hold promise as a viable approach to treat traumatic neuroma and prevent its development following peripheral nerve injury. Tissue-engineered materials offer several unique advantages in this context: (1) Firstly, biocompatibility and regenerative potential: Tissue-engineered materials are designed to be biocompatible, providing a suitable environment for cell adhesion, proliferation, and differentiation. This enhances the regenerative potential of the injured nerve tissue, promoting the formation of functional nerve fibers [12]. (2) Controlled drug delivery: Incorporating drug delivery systems into tissue-engineered constructs allows for the controlled release of therapeutic agents. This can help manage pain and inflammation, crucial aspects in the treatment of traumatic neuromas [13]. (3) Structural support and guidance: Tissue-engineered nerve repair materials can provide structural support and protection for the injured stump [14]. This is particularly important in preventing the formation of disorganized and painful neuromas. (4) Modulation of inflammatory response: Tissue-engineered materials can be designed to modulate the inflammatory response, reducing scar tissue formation and creating an environment conducive to nerve regeneration [15]. (5) Minimization of neuroma formation: The controlled environment provided by tissue-engineered constructs helps minimize the formation of painful neuromas, preventing the entrapment of regenerating nerve fibers within scar tissue [14].

This review first systematically summarizes previous studies on the formation of neuroma after peripheral nerve injury and traditional treatment methods in order to identify some challenges and future development directions of traditional neuroma treatment. Then, we focus on the latest research progress of clinical medicine and tissue engineering in the treatment of neuromas, mainly including the design and construction of tissue-engineered materials and the therapeutic effect. Based on the excellent research results of the latest treatment methods of neuroma in recent years, we will give the most promising treatment strategies in the future in order to promote the continuous improvement of neurological treatment effects.

## 2. Peripheral Nerve Injury, Regeneration, and Traumatic Neuroma Formation

### 2.1. Peripheral Nerve Injuries and Neuroma

Peripheral nerve injuries (Figure 1a) are commonly categorized into three main types based on the extent and severity of the damage [16]: (1) neuropraxia, (2) axonotmesis, and (3) neurotmesis. Each type has specific implications for the development of neuromas.

Neuropraxia (Figure 1(b-1)) represents the mildest form of peripheral nerve injury, typically resulting from compression or minor traction injury. It is characterized by reversible conduction block, where the structure of the axon remains intact, and only segmental reversible demyelination occurs. This type of injury is unlikely to lead to a neuroma as the structural integrity of the nerve remains intact [17]. However, the risk of neuroma increases if prolonged compression or traction of the nerve induces a persistent inflammatory response in the area, resulting in a disorder of fibroblast proliferation.

Axonotmesis, as illustrated in Figure 1(b-2), typically results from more severe trauma, frequently induced by crush injuries. It involves the interruption or severe destruction of axons following peripheral nerve injury, with antegrade degeneration potentially occurring in the distal segment of the injury. However, the surrounding supporting structures, particularly the endoneurium, remain intact [18]. During regeneration, axons endeavor to grow across the injury site, guided by the endoneurial canal. The type of axonotmesis that can cause neuroma formation is typically the more severe cases where there is significant damage to the axon and surrounding tissue. In such instances, the regenerative process may become dysregulated, resulting in the formation of a disorganized scar mass composed of fibroblasts, new nerve tissue, SCs, and other cellular elements.

Neurotmesis is the most severe form of peripheral nerve injury, involving complete transection of the nerve. This results in the loss of both axonal and connective tissue continuity. This type of peripheral nerve injury carries a heightened risk of neuroma formation, particularly in patients with complete transection (Figure 1(b-3)) and amputations (Figure 1(b-4)) [19].

Sunderland et al., based on the neuroma’s location, gross morphology, and damage pattern of nerve microstructure, systematically categorized neuromas into three groups [20,21]: Neuroma-in-continuity (Figure 1(c-1)) constitute 60–70% of the total number of neuromas. The distinctive structural feature of continuous neuromas is the presence of macroscopic expansion areas on intact nerve fibers. In most cases, the surrounding structures and internal supporting structures of the fibers remain intact, with lesions occurring on the internal axons. Although regenerated axons breach the constraints of the endoneurium, they are still shielded by the epineurium. Terminal or amputation neuroma (Figure 1(c-2)) share a similar nature, with nerve rupture being the direct cause of this type of neuroma. The distinction lies in the more pronounced clinical symptoms of the latter.

### 2.2. Peripheral Nerve Regeneration and Neuroma

Following peripheral nerve injury, Wallerian degeneration occurs (Figure 2a), where the axons distal to the stump undergo changes and disintegrate from proximal to distal due to the absence of axoplasmic transport, which is crucial for providing the nutrients needed to maintain and renew the axons [22]. SCs and macrophages phagocytose the disintegrated axons and myelin sheathing. In the regeneration stage, SCs proliferate first, forming the “Büngner bands” that act as a bridge for the regenerating branch bud of the proximal axon to reach the distal end of the fracture [23]. However, if this process is delayed or interrupted, debris accumulation will impede the formation of the Büngner bands, resulting in an abnormal immune microenvironment at the nerve injury site and an increased risk of traumatic neuroma formation.

The activation of SCs is crucial for guiding regenerating axons through the injury site (Figure 2b). Following peripheral nerve injury, SCs undergo dedifferentiation into a state with plasticity and regenerative capacity (reparative SCs), participating in Wallerian degeneration, Büngner band formation, and the secretion of neurotrophic factors [24]. Inadequate or excessive SC activity can result in aberrant axonal sprouting and branching, a common characteristic of traumatic neuromas. Factors such as the ratio of SCs to regenerating axons, the release of growth-promoting factors, and the maintenance of a permissive extracellular matrix collectively influence the outcome of nerve regeneration and the potential for neuroma formation.

Neurotrophic factors (Figure 2c) play a pivotal role in guiding axonal growth during nerve regeneration [13]. Insufficient or imbalanced availability of these factors can lead to misguided axonal sprouting and contribute to neuroma formation. Conversely, the excessive expression of the nerve growth factor can also transform it into a pain mediator, stimulating damaged nerve terminals and inducing hyperalgesia.

The relative balance of the immune microenvironment (Figure 2d) mediated by macrophages also plays a crucial role in the effective regeneration of peripheral nerves [25]. Chronic inflammation can directly impede the process of “neogenesis metabolism” in the peripheral nervous system. A significant number of newly formed nerve fibers cannot undergo myelination, and non-myelinated or partially myelinated nerve axons themselves are more susceptible to physical and chemical stimulation [26]. Additionally, as a consequence of excessive inflammatory response, inflammatory factors persist at the injury site for an extended period, leading to the repair of peripheral nerve injury remaining at the granulation stage. The formation of scar tissue (Figure 2c) and fibrosis at the injury site can impede proper nerve regeneration. Scar tissue may act as a physical barrier, preventing regenerating axons from reaching their target tissues. Additionally, the microenvironment created by scar tissue may promote aberrant axonal growth, increasing the risk of traumatic neuroma formation [27].

### 2.3. Mechanisms of Neuroma Formation

Morphologically, neuromas typically appear as bulbous structures attached to the proximal end of the nerve. Histologically, neuromas are non-neoplastic lesions characterized by a fibrous mass composed of fibroblasts surrounded by numerous axons, SCs, endothelial cells, and perifascicular cells [2]. They display the following features: (1) Compared to normal regenerating nerve fibers, there is a significant increase in demyelination or incomplete myelination of regenerated nerve fibers at the neuroma site [28]. (2) The regenerated nerve fibers are surrounded by a large number of disorganized fibroblasts [29]. (3) Early-stage tissue contains more mucous tissue glycosaminoglycans, while late-stage mature neuromas tissue contains more connective tissue and collagen. (4) An abnormal local immune microenvironment is present, characterized by an abnormal increase in multinucleated macrophages and excessive proliferation and differentiation of fibroblasts in a chronic inflammatory environment [30]. (5) The epineurium, perineurium, and endoneurium undergo thickening and irregular shape changes.

Studies have pinpointed crucial factors, including nerve growth factor (NGF), brain-derived neurotrophic factor (BDNF), and various cytokines, as key players in the formation and maintenance of neuromas. Animal experiments revealed that [31], after sciatic nerve transection in rats, all rats in the tropomyosin receptor kinase B (TrkB)-deficient group formed neuromas, while almost no neuromas were formed in the wild-type group. As a neurotrophic factor regulating neuronal cell survival, proliferation, and differentiation after peripheral nerve injury, the abnormal secretion of NGF at the injury site also plays a crucial role in the formation of painful neuroma and hyperalgesia. Studies [32] have demonstrated that the early inhibition of NGF and its surface receptor tropomyosin receptor kinase A (TrkA) after peripheral nerve injury can decrease the formation of painful neuromas and alleviate the pain caused by them. The significant findings related to confirming or denying the putative influences of traumatic neuroma are summarized in Table 1.

Studies [33] indicate that peripheral nerve injury induces a local inflammatory response and sensitization of nociceptors, leading to changes in conduction and increased transmission of pain impulses to the central nervous system. This alteration also affects the sympathetic nervous system, causing abnormal pain and resulting in discomfort. Furthermore, during peripheral nerve injury, Na^+^ channels on the nerve cell membrane open, allowing for a substantial influx of Na^+^ into the nerve cell. This stimulates immune cells to release various neuroinflammatory peptides (such as histamine, 5-hydroxytryptamine, substance P, calcitonin gene-related peptide, etc.) and induces abnormal electrophysiological activities. Consequently, the neuroma becomes excessively sensitive to various intensity stimuli in the periphery, leading to spontaneous discharge. This abnormal activity persists, sending ectopic impulses to spinal cord neurons, causing sensitization and abnormal sensory function at the spinal cord level, known as central sensitization.

The abnormal immune microenvironment at the site of peripheral nerve injury induces structural abnormalities in the developing peripheral nerve, resulting in abnormal increases in remyelination and demyelination [34]. Serving as the structure most closely associated with nerve axons, one of its functions is to protect axons. Non-myelinated or partially myelinated nerve axons, compared to normal nerve axons, exhibit increased sensitivity to physical and chemical stimulation. Furthermore, the myelin sheath plays a role in ensuring that, during the conduction of nerve impulses, nerve fibers remain isolated from each other. Loss of the myelin sheath eliminates this insulating state, allowing normal nerve conduction to directly jump to adjacent nerve fibers, causing ectopic conduction and resulting in abnormal itching and pain.

As another consequence of the excessive inflammatory response, the long-term presence of inflammatory factors at the injury site leads to the repair of peripheral nerve injury remaining at the granulation stage, and the expression of α-smooth muscle actin (α-SMA) is abnormally increased [35]. During the tissue healing process, fibroblasts resulting from chronic inflammation persistently proliferate and differentiate into myofibroblasts, continuously secreting a significant amount of collagen, leading to scar tissue deposition. The surplus collagen scar tissue exerts direct physical compression on the damaged nerve fibers, subjecting the typically sensitive nerve axons to additional physical stimulation. Simultaneously, α-SMA, acting as a spontaneous contractile protein, can induce persistent pain in neuromas without apparent external stimulation.

**Table 1 biomolecules-14-00484-t001:** A summary of pertinent research on established or presumed factors influencing traumatic neuroma formation, encompassing research methods, sources, research content, and conclusions.

Factors	Study and Methods	Source	Observations and Conclusions
NGF	The expression of NGF was quantified by immunohistochemistry [36].	Damaged nerve axons	The expression of NGF was significantly elevated in patients with traumatic neuroma compared to normal individuals
	The NGF receptors p75 and trkA and the BDNF receptor GFRalpha-1 were semiquantitatively analyzed by immunohistochemistry [37].	Glial cells	The immunoreactivity of trkA receptor was significantly increased in the traumatic neuroma group
	A mouse model of severe limb injury was used to study the role of sensory nerve fibers in fibrous scar tissue travel [38].	Myofibroblasts	NGF expressed by myofibroblasts is an important factor in the formation of neuroma after severe limb trauma
BDNF	Expression of BDNF and its receptor trkB after sciatic nerve transection in wild-type and heterozygous trkB-deficient mice [31].	Injured peripheral nerves	Trkb-deficient mice did not develop traumatic neuromas after long-stage peripheral nerve injury
	After sciatic nerve transection, the rats were given BDNF-containing and antagonistic BDNF-containing connective tissue chambers, respectively, and the incidence of traumatic neuroma was observed [39].	Endogenous secretion	BDNF plays a key role in the development of neuropathic pain after peripheral nerve injury, and its local inactivation reduces the incidence and severity of neuroma formation
	The formation of a neuroma 2 weeks after complete transection of the inferior alveolar nerve was confirmed by histological analysis [40].	Damaged nerve axons	Local administration of DNF antibody inhibited the proliferation of connective tissue at the injured site, promoted the integrity of nerve fibers, and reduced the formation of traumatic neuroma
Neuroinflammatory peptides	IL-6 antiserum or CGRP receptor antagonist was administered at the sciatic nerve ligation site [41].	Calcitonin gene-related peptide (CGRP) is expressed by the axons of neuromas	CGRP is involved in the formation of neuroma by upregulating the secretion of IL-6 in macrophages
	Neuroma model animals were treated daily with histidine and loratadine [42].	Histamine; mast cell	Endogenous histamine reduces neuropathic pain caused by traumatic neuromas
	Immunostaining was used to compare the contents of substance P and CGRP in axons of normal and neuroma sides [33].	Substance P; mast cell	The neuroma showed a large number of disorganized axonal contours and positive immunostaining for CGRP or Substance P
Inflammatory factor	To compare the expression of anti-inflammatory factors and proinflammatory factors in neuroma [43].	TNF-α, IL-1β, IL-6, and IL-10	IL-6 and IL-1β may play a role in the formation of traumatic neuroma, while IL-10 may inhibit neuroma formation
	The expression of inflammatory factor genes in the injured sciatic nerve was detected [44].	IL-1β, IL-10, IFN-γ, and TNF-α; macrophage	The high expression of proinflammatory and anti-inflammatory cytokines may be associated with the formation of fibrosis caused by irreversible nerve injury and, therefore, may be associated with the formation of traumatic neuroma
	Quantitative analysis of the expression of inflammatory factors in peripheral nerve stump [30].	TNF-α, IL-6, and IL-1β	Expression of proinflammatory cytokines TNF-α, IL-6, and IL-1β was significantly increased in the dorsal root ganglia of traumatic neuromas
α-SMA	The expression of α-SMA in neuroma was observed by immunofluorescence staining [35].	Myofibroblasts	The expression of α-SMA was positively correlated with the pain index of patients
	A rat model of amputated neuroma was used to quantitatively analyze the expression of α-SMA [45].	Myofibroblasts	Levels of α-SMA and the pain marker c-fos were significantly higher in the amputation group
	Quantitative analysis of α-SMA expression in the terminal of neuroma was performed [46].	Myofibroblasts	Capping transected rat sciatic nerves while concurrently administering myelin-associated glycoprotein was linked to reduced levels of α-SMA and autotomy behavior

## 3. Traditional Treatment Options and Challenges

### 3.1. Nonsurgical Treatments

Nonsurgical treatments for traumatic neuromas typically involve a multimodal approach, including pharmacotherapy and physical interventions. Medications such as tricyclic antidepressants and antiepileptic drugs may help manage neuropathic pain, while physical therapies, such as transcutaneous electrical nerve stimulation (TENS) or acupuncture, aim to alleviate symptoms and improve function. Pharmacological interventions represent a crucial conservative treatment, involving drugs with diverse mechanisms of action—specifically opioids, anticonvulsants, antidepressants, local anesthetics, steroids, and others (e.g., baclofen, Botox, ketamine, etc.)—that have shown efficacy in alleviating neuropathic pain associated with neuromas [47]. Opioids have a longstanding history of use in treating neural-related pain. They alleviate pain induced by neuromas by inhibiting pain-related signaling pathways in the spinal cord or reducing the activity intensity of the central nervous system, particularly pain intensity perception [48]. Clinical studies [49] have demonstrating that the intravenous, oral, or topical administration of opioids significantly alleviates pain caused by painful neuromas in amputee patients. To avoid respiratory depression, immunosuppression, and severe addiction caused by opioids, according to the Centers for Disease Control and Prevention (CDC) guidelines for chronic pain management with prescription opioids, medication protocols should adhere to principles such as short-term use, low initial doses, and standardized testing. Antidepressants, like amitriptyline, are occasionally used in managing traumatic neuromas because of their analgesic properties and ability to modulate neuropathic pain [50]. These medications can alleviate symptoms associated with the condition, such as pain and discomfort. However, their use may be associated with side effects including gastrointestinal disturbances, dry mouth, drowsiness, and potential interactions with other drugs. The topical application of steroid drugs like triamcinolone acetonide has an anti-inflammatory effect, significantly reducing the release of inflammatory mediators and thereby diminishing the stimulation of nerve axons [51]. Steroid drugs have an immunosuppressive effect, which reduces the demyelination of nerve fibers, abnormal power generation, and inhibits excessive disordered growth of nerve fibers. It also inhibits scar formation, reduces fibrocyte infiltration, and enables nerve fibers to pass through the junction site smoothly [52]. Local anesthetics like lidocaine not only relieve pain but also act as sodium channel blockers, inhibiting the abnormal discharge of primary sensory neurons and fibers to reduce the stimulation of the central nervous system. Local anesthetic and/or steroid injections along painful neuromas may provide pain relief, although the results are usually temporary [53].

### 3.2. Surgical Treatments

The selection of a surgical procedure for neuromas should be based on the type of injury. While resection and end-to-end anastomosis theoretically offer the potential to restore neural continuity, reinstate sensorimotor function, and alleviate symptoms, this approach is notably restricted for neuroma-in-continuity cases [54]. In practice, the substantial extent of neuroma resection often results in a large nerve gap, leading to tension during end-to-end nerve anastomosis. Consequently, this surgical method is rarely employed in clinical practice. In such cases, autograft and allograft emerge as more suitable and commonly utilized treatments [11]. However, the limited availability of human nerve tissue and the collection process unavoidably lead to permanent loss of nerve transmission function at the donor site, along with some associated complications. Therefore, during nerve harvesting, the principle of repairing more critical nerves with less critical nerves is often followed [55].

For patients with amputation-type terminal neuromas, simple neuroma resection is a less intricate procedure. However, the significant limitation of this surgical approach is its inability to prevent the recurrence of neuromas at the excision site. Although reliable large-sample statistical data are still lacking, surgeons have consistently warned about its high recurrence rate [56]. After all, the microenvironment leading to neuroma formation remains unchanged with simple physical resection. Subsequently, in addition to neuroma resection, nerve stump decompression has been further developed. This involves burying truncated healthy nerve fibers in nearby nutrient-rich muscle tissue, a bone marrow cavity, or blood vessels [57]. This technique buries the nerve endings deep into the tissue, providing additional buffering and protection for the nerve, and has shown some efficacy in preventing the formation of traumatic neuromas. However, this technology does not fully address the issue.

Targeted muscle reinnervation (TMR) [58] and regenerative peripheral nerve interfaces (RPNIs) [59] are technologies that have been developed in recent years through advances in biomedical engineering, neurology, and regenerative medicine. These technologies aim to achieve the effective re-innervation of target organs. TMR involves surgically transferring the residual nerves from an amputated limb to nearby target muscles. This redirects the nerve endings to innervate the muscle tissue, providing a pathway for the axons to grow and preventing the formation of neuromas at the nerve ending sites. By integrating the nerves into functional muscle tissue, TMR offers a more natural environment for nerve regeneration, reducing the risk of neuroma formation [27]. Similarly, RPNIs create a conduit for nerve regeneration by connecting a severed nerve to a nearby muscle or other suitable tissue. This redirects the regenerating axons away from the site of injury, dispersing them into the muscle tissue and minimizing the formation of neuromas. By providing a permissive environment for nerve regeneration, RPNIs promote more organized nerve growth and reduce the likelihood of neuroma development.

Addressing the aforementioned shortcomings in neuroma treatment involves achieving the effective bridging of nerve transection injuries for the reinnervation of distal target organs. Additionally, it entails ensuring the effective protection of the nerve stump to isolate surrounding tissues and counteract the effects of influencing factors and inflammatory elements that promote neuroma growth. This represents the direction for improving painful neuroma treatment. Recent advancements in biomedical materials, rooted in tissue regenerative engineering, have significantly progressed in the prevention and treatment of neuroma. The advantages of tissue-engineered peripheral nerve repair materials over conventional treatment options are summarized in Table 2. Personalized implantable medical materials, designed to facilitate effective functional tissue regeneration, have garnered the interest of researchers and clinicians for their substantial potential in preventing and treating painful neuromas [60]. Various materials, such as autologous vein, acellular nerve matrix, chitosan, gelatin, and synthetic polymer materials, have been employed to protect the nerve stump [61].

## 4. Tissue-Engineered Materials for Neuromas

Traditional treatment methods for neuromas, including pharmacological interventions, surgical resection, implantation of the proximal nerve stump into adjacent tissue, or TMR, can only relieve pain; these methods do not fundamentally address the underlying problem of incorrect repair following peripheral nerve injury. Tissue-engineered materials leverage the principles of regenerative medicine, aiming to create a conducive microenvironment for nerve regeneration rather than solely focusing on symptomatic relief. This shift in paradigm addresses the root cause of nerve injury, offering a more comprehensive solution. Tissue-engineered strategies for traumatic neuroma treatment involve a multi-faceted approach, encompassing the use of biomaterials, growth factors, and cellular components.

### 4.1. Composition and Types of Tissue-Engineered Materials

#### 4.1.1. Composition of Tissue-Engineered Materials

Peripheral nerve tissue engineering scaffolds serve as templates to guide and support nerve regeneration. These scaffolds can be broadly classified into natural and synthetic materials, each with distinct properties and applications. Natural materials sourced from biological origins offer excellent biocompatibility and bioactivity, facilitating cell attachment, proliferation, and differentiation [27]. The most representative of these are chitosan and gelatin. Chitosan, derived from chitin, exhibits favorable biocompatibility, biodegradability, and antimicrobial properties [62]. The chitosan-based nerve repair scaffold has good mechanical and mechanical performance, which can protect the injured peripheral nerves and provide good structural support for them. Its excellent biocompatibility is conducive to cell adhesion and proliferation and provides a suitable microenvironment for peripheral nerve regeneration and repair. Collagen, a major structural protein in the extracellular matrix, is widely used in nerve tissue engineering due to its biocompatibility, biodegradability, and resemblance to native nerve tissue [63]. Collagen-based scaffolds promote cell adhesion and migration, supporting nerve regeneration. However, rapid degradation and mechanical weakness are significant limitations.

Poly(lactic-co-glycolic acid) (PLGA) and poly(caprolactone) (PCL) are the most widely used synthetic materials [64]. The advantage of PLGA and PCL is that they can precisely control scaffold properties, including mechanical strength, degradation rate, and pore size distribution, enabling tailored designs for nerve regeneration. PLGA is a biodegradable copolymer widely used in tissue engineering due to its excellent mechanical properties and controllable degradation rate [65]. PLGA scaffolds provide structural support and the controlled release of bioactive agents, promoting nerve regeneration. However, acidic degradation byproducts may induce inflammation, limiting its application. PCL is a biocompatible and biodegradable polyester with adjustable mechanical properties. PCL scaffolds maintain structural integrity during nerve regeneration and degrade slowly, providing long-term support [66]. Nevertheless, PCL lacks inherent bioactivity, necessitating surface modifications to enhance cell interactions.

#### 4.1.2. Type of Tissue-Engineered Materials

At present, tissue-engineered materials for peripheral nerve repair mainly include nerve guide conduits, hydrogels, electrospinning fibers, and microsphere, etc., which promote nerve regeneration after peripheral nerve injury, provide a good regeneration microenvironment, and prevent the formation of traumatic neuroma [67].

Nerve guide conduits (Figure 3a) are tubular structures designed to bridge nerve gaps and guide axonal regeneration, offering several advantages in peripheral nerve repair [62]. Conduits provide a confined environment for nerve regeneration, facilitating directional growth and preventing aberrant sprouting. Conduits made from biodegradable materials minimize immune responses and promote tissue integration, avoid the abnormal proliferation and differentiation of fibroblasts caused by immune abnormalities at the nerve injury site, and reduce the formation of traumatic neuroma [68]. Conduits shield regenerating nerves from surrounding tissues, reducing the risk of compression or secondary injuries, and provide structural support for peripheral nerve regeneration.

Hydrogels (Figure 3b), water-swollen polymer networks with unique properties, emerge as promising candidates for peripheral nerve repair applications [69]. Mimicking the hydrated extracellular matrix, hydrogels promote cell adhesion, proliferation, and differentiation [70]. Hydrogels have the capability to encapsulate growth factors or cells, enabling sustained release and localized delivery to the injury site, thus promoting nerve regeneration. Injectable hydrogels conform to irregular defect geometries, allowing for minimally invasive delivery and enhancing surgical precision and patient comfort [71].

Electrospinning (Figure 3c) is a versatile technique used to fabricate nanofibrous scaffolds, offering unique advantages for peripheral nerve repair [72]. Electrospun fibers, with their high surface-area-to-volume ratio, promote cell adhesion, proliferation, and neurite outgrowth. Fiber alignment can be precisely controlled to mimic the native nerve architecture, facilitating directional axonal growth [73]. Electrospun fibers can be reinforced with polymers to enhance mechanical strength and stability, thereby providing structural support during nerve regeneration.

Microspheres (Figure 3d) provide a distinctive approach for delivering bioactive molecules or cells to facilitate nerve repair [74]. They can encapsulate growth factors, cytokines, or cells, enabling controlled release and sustained delivery to the injury site. The localized release of neurotrophic factors from microspheres enhances neuronal survival, axonal growth, and myelination, thereby promoting nerve regeneration. Microspheres can be incorporated into scaffolds or injected directly into the injury site, thereby facilitating scaffold integration and enhancing therapeutic efficacy [75].

### 4.2. Biomaterial-Based Scaffolds

Recognizing the intimate connection between prolonged chronic inflammation following peripheral nerve injury and the development of traumatic neuroma, a nerve guide conduits endowed with the capability to regulate the immune microenvironment using a modified polydopamine coating approach is constructure [15]. The polydopamine coating, modified on the surface, markedly enhances the hydrophilicity of nerve repair materials, fostering the adhesion, proliferation, and differentiation of Schwann cells. In the 10 mm sciatic nerve defect model of rats, it exhibited a therapeutic efficacy comparable to autologous nerve transplantation in promoting peripheral nerve regeneration. Importantly, the use of this nerve repair material in the animal model of terminal neuroma markedly decreased macrophage accumulation and the secretion of inflammatory factors at the nerve injury endpoint, leading to a reduction in terminal painful neuroma formation.

The application of nerve repair material capping shows potential to prevent terminal or amputation neuroma (Figure 4a–c). Yi et al. conducted a synthesis of nerve repair material using L-polylactide (PLLA) and modified its surface with arginylglycylaspartic acid (RGD peptide) to hinder the formation of terminal neuromas [76]. This material inhibits the accumulation of T cells and macrophages at the nerve injury site, suppresses the expression of genes associated with inflammatory factors, reduces collagen deposition, and hinders scar formation. Moreover, it promotes the remyelination of injured nerves, with the goal of averting the development of traumatic neuroma.

Preventing neuroma formation involves creating a barrier over the injured peripheral nerves to inhibit aberrant axonal growth. Utilizing bio-materials such as extracellular matrix components or bioengineered scaffolds can effectively cap the injured nerves [77]. These biomaterials offer a supportive environment for regenerating axons while preventing the formation of neuromas by inhibiting the disordered growth of regenerated nerve fibers and promoting the myelination of injured nerves. Additionally, incorporating nerve growth factors, nerve regeneration drugs, or stem cells into these materials can modulate local immune-inflammatory responses, inhibit myofibroblast invasion, and enhance nerve regeneration potential, thereby reducing neuroma formation. Qiu et al. sought to deter neuroma formation by utilizing acellular neural matrix scaffolds [12]. Research indicates that this scaffold, mimicking the microstructure of a natural nerve, can effectively guide axon regeneration and their orderly arrangement. Furthermore, its adventitia–adventitia barrier prevents the invasion of vascular muscle scar tissue, significantly reducing fibroblast infiltration into the nerve stump, thereby effectively inhibiting scar formation and guiding the nerve stump’s gradual transformation into benign tissue. This makes it an exceptional nerve repair scaffold for treating neuroma. Biocompatible hydroxyethyl cellulose (HEC)/soy protein isolate (SPI) sponge capping conduits (HSSCs) were developed to prevent the formation of traumatic neuromas [78]. In vivo, after capping the sciatic nerve stump, the capped group exhibited a significant reduction in inflammatory invasion and collagen overdeposition compared to the uncapped group. Additionally, the expression of genes related to remyelination in the nerve terminal was twice as high in the capped group as in the uncapped group.

Equipping the nerve repair scaffold with a directional internal structure to guide SCs and axons towards regeneration in the distal target organ, incorporating a porous structure to facilitate material exchange inside and outside the nerve conduit, and ensuring excellent biocompatibility to create a conducive microenvironment for nerve regeneration are effective strategies to prevent nerve transection injuries from progressing into traumatic neuromas (Figure 4d–f). Millán et al. developed a porous nerve repair scaffold with directional channels using type I collagen, an excellent biocompatible material [79]. This design aimed to achieve the swift repair and precise docking of 10 mm peripheral nerve defects, thereby preventing the formation of traumatic neuromas. In vitro studies demonstrated that directional channels could steer the growth of regenerated axons and promote the early adhesion and directional proliferation of Schwann cells. In a 10 mm sciatic nerve defect animal model, motor function and peripheral nerve regeneration attained results comparable to autologous nerve transplantation, significantly reducing the risk of traumatic neuroma.

**Figure 4 biomolecules-14-00484-f004:**
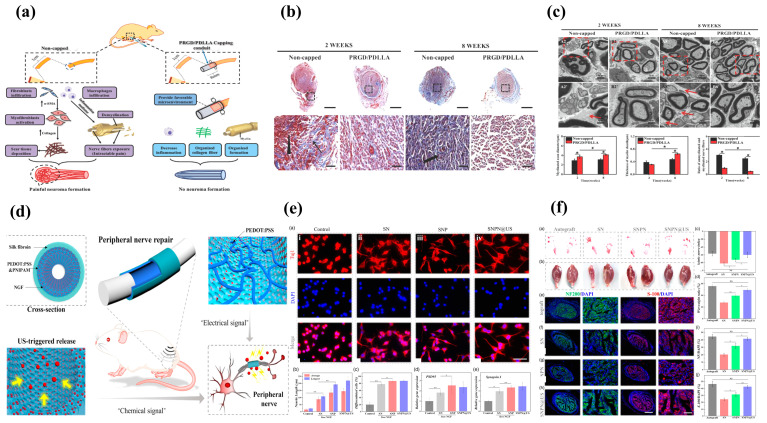
Tissue-engineered materials were employed on the amputated nerve stump to cap the injured peripheral nerve (**a**), thereby reducing heightened myofibroblast differentiation (**b**) and significantly enhancing the remyelination of regenerated axons. (**c**) Copyright (2018) Wiley [76]. In cases of nerve transection injuries (**d**), the nerve repair scaffold can stimulate the proliferation and differentiation of SCs (**e**), guiding and bridging the injured peripheral nerves for swift and precise growth into the distal target organs. (**f**) Copyright (2023) Wiley [80]. Fundamentally, tissue-engineered materials can prevent the occurrence of neuroma after peripheral nerve injury. In the picture, *, **, #, NS represents statistically significant difference (*p* < 0.05).

### 4.3. Growth Factor Incorporation

The use of growth factors in the treatment of traumatic neuroma holds promise as a potential therapeutic approach. Growth factors are signaling molecules that play crucial roles in various cellular processes, including cell proliferation, differentiation, and migration [13]. In the context of traumatic neuromas, the application of growth factors aims to modulate the regenerative environment, promoting organized axonal growth and reducing the likelihood of disorganized, aberrant sprouting associated with neuroma formation. In the context of tissue-engineered approaches for traumatic neuroma treatment, growth factors are often encapsulated within biomaterial scaffolds. This controlled release of growth factors from the engineered constructs ensures a sustained and localized delivery, optimizing their therapeutic impact over time. This is particularly important for maintaining the regenerative signals throughout the critical phases of nerve healing, and the currently applied nerve growth factors for peripheral nerve repair are summarized in Table 3.

Growth factors such as nerve growth factor (NGF), brain-derived neurotrophic factor (BDNF), and glial cell line-derived neurotrophic factor (GDNF) have demonstrated the ability to promote neurite outgrowth. By enhancing the extension of regenerating axons, these growth factors may contribute to more targeted and organized reinnervation, reducing the risk of neuroma formation. Xu et al., infused the PLLA/β-tricalcium phosphate (β-TCP) nerve conduit with chitosan (CS)–hyaluronic acid (HA) hydrogel loaded with NGF to achieve the sustained release of growth factors at the site of peripheral nerve injury [81]. In vitro studies have demonstrated that this sustained release of NGF significantly enhances neuronal cell adhesion, proliferation, and differentiation. In a rat model of long-segment (10 mm) sciatic nerve defect, the nerve repair conduit with continuous NGF release markedly enhances the re-innervation of target organs and myelination of regenerated axons, effectively preventing the formation of persistent neuroma.

Growth factors can act as guidance cues for regenerating axons. They help create a supportive microenvironment by influencing the orientation and direction of axonal growth. This guidance is crucial for preventing misdirection and inappropriate branching, common features of traumatic neuromas. Growth factors work in tandem with Schwann cells to guide axons across the injury site. Zhu et al. engineered an aligned poly(ε-caprolactone) (PCL) fibrous conduit featuring an NGF concentration gradient to achieve the rapid and precise repair of a 15 mm peripheral nerve defect [82]. Research indicates that the NGF concentration gradient exerts a chemotactic effect on regenerated axons, guiding their directional regeneration from a low to high concentration. This tissue-engineered material, fostering guided axonal regeneration, proves effective in preventing the denervation of distal target organs and significantly reducing the risk of neuroma.

**Table 3 biomolecules-14-00484-t003:** Summary of the receptors, molecular signaling pathways, and bio-functions of growth factors employed to facilitate rapid nerve regeneration after peripheral nerve injury, aiming to prevent traumatic neuroma.

Growth Factors	Receptors	Signaling Pathways	Bio-Function in Promoting Peripheral Nerve Regeneration
Nerve growth factor (NGF) [83]	TrkA, p75^NTR^	PI3K/Akt MAPK/ERK PKC signaling	Guided the directional regeneration of nerve fibers; Promote the proliferation and differentiation of SCs; Regulate the immune microenvironment; Avoid injured peripheral nerves apoptosis and prevent denervation.
Brain-derived neurotrophic factor (BDNF) [84]	TrkB, p75^NTR^	PI3K/Akt MAPK/ERK JAK/STAT	Protect the injured peripheral nerve cells; Proliferation, migration, and promyelination of SCs.
Glial-derived neurotrophic factor (GDNF) [85]	GFRα 1~4	PI3K/Akt MAPK/ERK	Repair and nutrition of injured peripheral nerves; Promoting synaptic differentiation and axon regeneration in peripheral neurons.
Neurotrophin-3 (NT-3) [86]	TrkC, p75^NTR^	PI3K/Akt MAPK/ERK JNK/c-Jun	Maintain the survival of neurons and promote their differentiation and proliferation; Induces axonal growth.
Insulin-like growth factor (IGF-1) [87]	IGF 1R, and 2R	PI3K/Akt MAPK/ERK JNK/c-Jun	Promotes the proliferation and differentiation of SCs into myelin; Promotes the directional growth of nerve fibers into the distal target organs; Regulation of the immune microenvironment.
Basic fibroblast growth factor (bFGF) [88]	FGFR I-IIIc	PI3K/Akt MAPK/ERK JAK/STAT	Regulates the proliferation and differentiation of fibroblasts; Promotes angiogenesis; Protect injured peripheral nerves and avoid denervation.
Vascular endothelial growth factor (VEGF) [89]	VEGF R1~R3 NP1, NP2	PI-3K/Akt MAPK/ERK	Alleviates ischemia and promotes recovery of blood supply after nervous system injury; Nutrition and protection of injured peripheral nerves.

### 4.4. Cell-Based Approaches

Tissue-engineered cell-based therapy provides a platform to guide and promote axonal regeneration (Figure 5a). Transplanted cells, such as SCs or stem cells, can be strategically placed within engineered scaffolds to facilitate the organized growth of axons, preventing the formation of disorganized and painful neuromas [90]. Biomaterial scaffolds in tissue engineering serve as a supportive microenvironment for transplanted cells. These scaffolds mimic the natural extracellular matrix and provide physical support, allowing transplanted cells to integrate into the host tissue and create an environment conducive to nerve regeneration (Figure 5b,c). Muangsanit et al. induced angiogenesis at the peripheral nerve injury site by incorporating aligned vascular endothelial cells into tissue-engineered hydrogels [91]. This approach establishes a favorable microenvironment for the regeneration of injured peripheral nerves, reducing the occurrence of misregulated neuroma. Our study illustrated that treatment with aligned tissue-engineered endothelial cells significantly enhanced axon regeneration and interspace vessel formation in a 10 mm defect in vivo, achieving therapeutic effects comparable to autologous nerve treatment.

Cells used in tissue engineering can be encapsulated within biomaterials, allowing for the controlled release of trophic factors and other signaling molecules. This controlled release ensures a sustained and localized therapeutic effect, optimizing the regenerative process over time [92]. Recognizing the pivotal roles of Schwann cells and vascularization in peripheral nerve regeneration (Figure 5d–f), Wu et al. genetically modified Schwann cells to overexpress vascular endothelial growth factor A (VEGF-A) and incorporated them into the inner wall of hydroxyethyl cellulose/soy protein isolate/polyaniline sponge (HSPS) conduits [93]. Research indicates that the sustained release of VEGF-A fosters Schwann cell proliferation, migration, and differentiation. In a 10 mm defect model of the sciatic nerve, this approach significantly enhances vascularization and nerve fiber regeneration at the injury site, holding the potential to replace autologous nerve grafting.

Tissue-engineered cell-based therapy often involves the combination of transplanted cells with growth factors. The synergistic effects of both components enhance nerve regeneration by promoting axonal growth, cell survival, and the modulation of the local microenvironment [94]. To address the limitation of SCs in forming Bungner bands to guide and bridge regenerating axons into target organs in peripheral nerve transection injury, Zhou et al. pre-cultivated a nanofiber nerve conduit with nerve growth factors and bone marrow stromal cells [95]. This approach aims to mitigate the risk of target organ loss during peripheral nerve regeneration across extended segments. The engineered scaffold, loaded with both growth factors and stem cells, notably enhanced the regeneration and remyelination of nerve fibers in a 15 mm segmental defect. Similarly, leveraging the remarkable efficacy of stem cells and growth factors in treating large-gap peripheral nerve injuries, Luo et al. developed a hydrogel loaded with a basic fibroblast growth factor (bFGF) and dental pulp stem cells (DPSCs) to provide accurate bridging guidance for such injuries [96]. In a rat model of a 15 mm long sciatic nerve defect, the collaborative action of stem cells and growth factors notably enhanced the regeneration of nerve fibers and contributed to the functional recovery of target organs.

**Figure 5 biomolecules-14-00484-f005:**
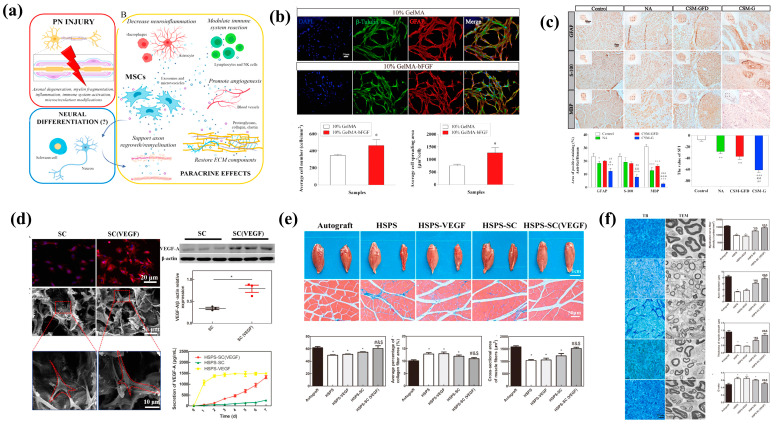
Cell therapy is an effective approach to promote the repair of peripheral nerve injuries and prevent the development of neuroma (**a**) [97]. Copyright (2021) MDPI. Tissue-engineered materials with excellent biological activity can be loaded with therapeutic stem cells (**b**) to address the challenges associated with the regeneration and repair of peripheral nerve transection injury (**c**) [96]. Copyright (2021) Elsevier. When combined with growth factors (**d**), cell therapy can create a favorable microenvironment for the regeneration of injured peripheral nerves, facilitating the re-innervation of target organs (**e**), remyelination of regenerated axons (**f**) [93], copyright (2021) Elsevier, and preventing the formation of traumatic neuroma. In the picture, *, **, ***, #, ##, ###, $, &, &&, &&& represents statistically significant difference (*p* < 0.05).

### 4.5. Electrical Stimulation

Incorporating electrical stimulation into tissue-engineered approaches has shown promise in enhancing nerve regeneration. Electrical cues can be integrated into the biomaterial scaffolds, providing a stimulating environment for regenerating axons and promoting their alignment. Electrical stimulation from bioelectrical active materials can enhance neuronal excitability in the injured nerve, facilitating the transmission of nerve impulses [98]. This is crucial for restoring proper functionality to the affected nerves and reducing symptoms such as pain and sensory abnormalities associated with trauma.

Bioelectrical active materials, such as conductive polymers and electrode-based devices, can provide electrical stimulation to injured nerves. Electrical stimulation has been demonstrated to promote axonal growth, guide regenerating axons, and enhance the overall rate of nerve regeneration. This advantage is particularly significant in cases of traumatic neuromas where the normal regenerative process may be compromised [99]. Song et al. developed a peripheral nerve repair conduit using the conductive polymer polypyrrole (PPy) and incorporated human neural progenitor cells (HNPCs) into the conductive nerve repair material for treating advanced-stage peripheral nerve injuries [100]. In vitro studies demonstrated that electrical stimulation significantly upregulated the gene expression of neurotrophic factors crucial for synaptic remodeling, nerve regeneration, and myelination in HNPCs. In a 10 mm rat sciatic nerve defect model, the regenerated axons and their remyelination exhibited satisfactory outcomes. Stem cell therapy enhanced by electrical stimulation proves to be an effective approach in preventing the formation of neuroma following extensive peripheral nerve injury. He et al. demonstrated the significant advantages of electrical stimulation in peripheral nerve regeneration through the utilization of bioelectrically active carbon-nanotube-doped conductive nerve repair materials [101]. In vitro 2D and 3D culture experiments revealed that electrical stimulation induced Schwann cells to migrate out of the dorsal root ganglion (DRG) to form “Bungner band”-like segments. Additionally, axonal outgrowth and myelination were significantly enhanced. However, the biocompatibility of traditional external electrical stimulation devices is not ideal, and their rigid structure imposes certain immune obligations on the surrounding tissues. This is not conducive to the repair and regeneration of peripheral nerves and, to some extent, diminishes their advantages in treating traumatic neuroma. Yang et al. developed a flexible, self-powered, and conductive scaffold using carbon nanotube@gelatin methacryloyl/poly (L-lactic acid) (CNTs@GelMA/PLLA) [102]. This scaffold aims to provide endogenous piezoelectric stimulation and a conductive microenvironment. In vitro quantitative data demonstrated that this composite scaffold significantly enhanced Schwann cell adhesion and elongation, promoted axon growth, and increased the number of axons in the dorsal root ganglion, thereby reducing the risk of neuroma formation in peripheral nerves.

Bioelectrical active materials can be engineered to enable the controlled release of neurotrophic factors. Neurotrophic factors play a key role in promoting neuronal survival, axonal growth, and the overall regenerative process. By integrating these factors into bioelectrical active materials, a sustained and localized release can be achieved, optimizing their therapeutic effects [103]. Zhang et al. developed an ultrasonically responsive composite conductive wire conduit utilizing a novel poly (3,4-ethylene dioxythiene)-polystyrene sulfonate (PEDOT:PSS) to achieve the controlled, slow release of NGF at the peripheral nerve injury site [80]. In a 10 mm nerve defect model, the combination of this conductive material with ultrasound-triggered NGF markedly enhanced nerve fiber regeneration and myelination. This represents a promising therapeutic approach for preventing neuroma formation.

## 5. Summary and Challenge

Neuroma is an abnormal proliferation of disorganized nerve tissue resulting from nerve injury, characterized by pain, hypersensitivity, and the formation of nodular structures. The regenerated nerve fibers grow disorderly, leading to the loss of distal target organs. Throughout the process of nerve regeneration, abnormalities in degeneration, SC activation, denervation and reinnervation, neurotrophic factors, scar tissue, and fibrosis contribute to neuroma formation. Current treatment options for traumatic neuroma include surgical interventions such as neuroma resection and nonsurgical approaches involving medications (antidepressants, anticonvulsants, opioid analgesics, etc.), massage, desensitization, radiofrequency ablation, etc. Tissue-engineered nerve regeneration materials offer a means for treating traumatic nerves and preventing neuroma formation. Therapeutic strategies, including nerve regeneration-guided scaffolds, nerve growth factor incorporation, cell therapy, and bioelectrical activity stimulation, can bridge peripheral nerve transection injuries or cap nerve stumps.

Current treatments for neuromas often provide only symptomatic relief without addressing the root cause. Tissue engineering represents a breakthrough, offering a more comprehensive solution by leveraging advanced biomaterials, growth factors, and cell-based approaches. However, challenges persist, including the need for optimal biomaterial selection, ensuring sustained growth factor release, and achieving seamless integration with host tissue. Shortcomings in mimicking the intricate microenvironment of nerve tissue pose hurdles in promoting ideal axonal regeneration. In the future, nerve repair scaffolds will be developed to be more in line with the physiological microenvironment of peripheral nerve regeneration and the three-dimensional structure of bionic autologous nerves. This aims to prevent and treat neuromas following peripheral nerve injury.

## Figures and Tables

**Figure 1 biomolecules-14-00484-f001:**
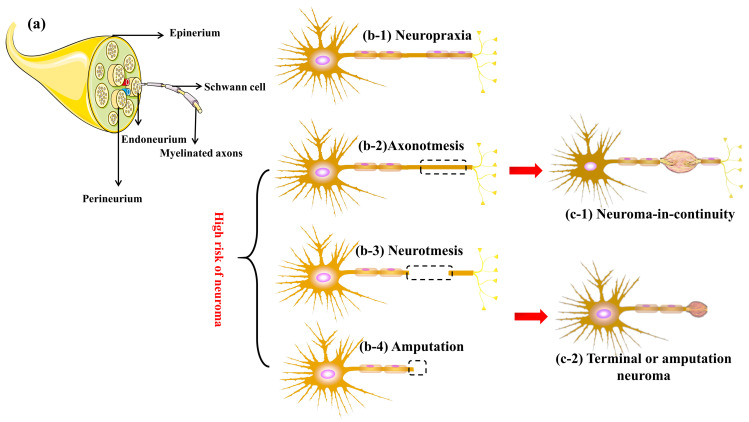
Schematic representation of neuroma formation. (**a**) Normal peripheral nerve anatomy. The risk of neuropraxia (**b**-**1**) developing into neuroma is low, while patients with axonotmesis (**b**-**2**), neurotmesis (**b**-**3**), and amputation (**b**-**4**) have a significantly increased risk of developing neuroma-in-continuity (**c**-**1**) and terminal neuroma (**c**-**2**).

**Figure 2 biomolecules-14-00484-f002:**
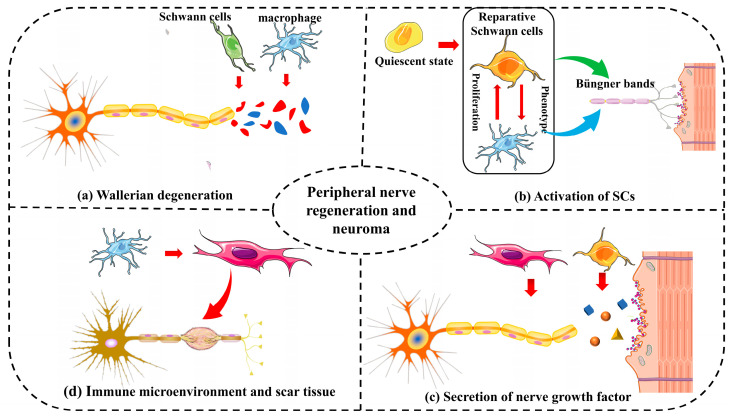
The process of peripheral nerve regeneration is intricately linked with the physiological course of traumatic nerve regeneration. Wallerian degeneration, orchestrated by SCs and macrophages, establishes a conducive microenvironment for axonal regeneration (**a**). M2 macrophages release various nerve growth factors (**c**), while Büngner bands, formed by SCs, guide axonal regeneration (**b**). An abnormal immune microenvironment, orchestrated by macrophages and their secreted inflammatory factors, induces aberrant proliferation and differentiation of fibroblasts, resulting in scar tissue formation and an elevated risk of traumatic neuroma (**d**).

**Figure 3 biomolecules-14-00484-f003:**
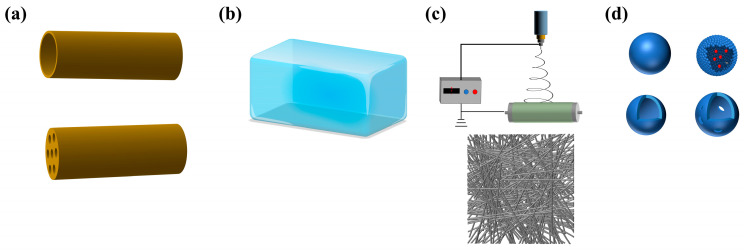
Tissue-engineered materials applied to peripheral nerve repair. (**a**) nerve guide conduits; (**b**) hydrogels; (**c**) electrospinning fibers; (**d**) microspheres.

**Table 2 biomolecules-14-00484-t002:** Summary of tissue-engineered materials alongside conventional nonsurgical and surgical treatment options, providing a comparison of their respective advantages and disadvantages.

Therapeutic Regimen	Methods	Objectives	Advantages	Disadvantages
Nonsurgical treatment	Pharmacotherapy; Radiofrequency ablation; Rehabilitation massage; Acupuncture; Electrical stimulation.	Relieve the pain symptoms of patients	Lower costs; The compliance of patients is high; Secondary injury can be avoided.	Cannot be treated fundamentally; Symptom relief is limited; Requires long-term repeated intervention.
Surgical treatment	Excision of neuroma; Nerve stump dredging; Epineurium and perineurium suture; Reinnervation.	Relieve the pain symptoms of patients; Remove the existing traumatic neuroma; Reinstate the continuity of peripheral nerves.	Cn remarkably relieve the symptoms of pain; The neuroma was completely removed; The nerve stump can be covered within the bone marrow cavity and other tissues to safeguard the nerve.	The recurrence rate is high; Limited by the anatomical site; The formation of neuroma cannot be prevented fundamentally.
Tissue-engineered materials	The injured nerve ends were capped for protection; Bridging the injured peripheral nerves.	To prevent the formation of neuromas; To promote the rapid and accurate docking of peripheral nerves.	The formation of neuroma after nerve injury is fundamentally solved; Individualized treatment; Precise treatment.	Development of a nerve repair material that aligns with the microenvironment of peripheral nerve regeneration, which requires further exploration

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
