# Peer review of "Strategies for Treating Traumatic Neuromas with Tissue-Engineered Materials"

_biomolecules, 2024, doi:10.3390/biom14040484_

Round 1

Reviewer 1 Report

Comments and Suggestions for Authors

The paper is well-written and covers an interesting topic on neural tissue damage recovery using tissue engineering. The references are appropriately recent.

Suggestions:

Title change: Strategies for treating traumatic neuromas with tissue engineering materials.

The authors get to the point (related to the manuscript title) only in chapter 5. Therefore, adding slightly more elementary information about materials (the type and composition) used for treating neural traumas would improve the Introduction section. For example, a bit more expanded statement, beginning in line 338 of the manuscript, if included in the Introduction, would improve the quality of the chapter. 

Chapters 2 and 3 should be shortened and fused.

Reviewer 2 Report

Comments and Suggestions for Authors

The term, “tissue engineering materials” may not be the best term grammatically. Perhaps ““tissue-engineered materials” would be more appropriate. The latter term is used at times in the manuscript and should probably replace the other, similar terms.

Line 62, delete “unsatisfactory”.

Line 65, “Tissue engineering materials holds…” should probably be Tissue-engineered materials hold…”

Line 86, “focuses” should be “focus”.

Line 109, “result” should be “results”.

Line 110, “They involve” should be “It involves”.

Line 116, “nerve gaps” occur in neurometic injuries but not axonometic injuries. Although you could use “length of degenerated axon” or something similar here instead, this is likely less predictive of neuroma formation that the amount of scar tissue that develops at the site of injury. It is this scar that impedes axonal pathfinding (thus creating the neuroma) rather than the length the axon needs to grow to reach its target. This is discussed in Figure 2.

IN the paragraph beginning with Line 118, neuroma-in-continuity is discussed in the section dealing with neurometic injuries. This might confise the reader. Perhaps it would be better to discuss this type of neuroma (which is not a transection injury) in the axonometic injury section in the paragraph prior. Similarly, please revise Figure 1 to show neuroma-in-continuity a type of axonometic injury and not a transection injury.

Line 185, “Prolonged denervation can result in irreversible changes, making it challenging for regenerating axons to find their original targets.” Is this true? Prolonged denervation results in irreversible changes in the muscle, so that even if reinnervation occurs, the muscle is too degenerated to function. This is a separate issue that has nothing to do with neuroma formation at the site of nerve injury, however.

Line 278, “The standard treatment for patients undergoing painful nerve resection involves opioid therapy for 4 to 6 weeks post-surgery.” This is incorrect.

Line 280, “surgeons now recommend precise local administration under ultrasound guidance”. This is incorrect. As a surgeon, I have never heard of USG-guided injection of an opioid, anywhere in the body much less along a nerve.

Section 4.1 would probably be better if you used the following pharmacologic subsections: opioids, anticonvulsants, antidepressants, local anesthetics, steroids, other (e.g. baclofen, Botox, ketamine, etc.). These are the medications used most commonly in the treatment of painful neuromas.

Line 296, “For neuroma-in-continuity resulting from long segments and destructive injuries of peripheral nerves…”. Maybe just say, “For neuroma-in-continuity…”. The rest is redundant. 

Line 311, why do you mention “tumor” here? Iti si probably the wrong term.

Line 318, “Although highly effective…” This technique is only modestly effective, not highly effective. Results can be quite disappointing, actually. The real issue with implanting nerve into non-denervated muscle, is that there is no biological reason for the axons to grow into and reinnervate the muscle. Thus, the neuroma reforms. In contrast, axons want to grow into devervated muscle instead. This is why tergeted muscle reinnervation (TMR) and regenerative peripheral nerve interface (RPNI) procedures were developed, so that the axons can grow into denervated muscle. Please include these procedures in this section. They represent the current standard of care for surgical management of stump neuromas.

Line 383, “Capping injured peripheral nerves to create a favorable microenvironment for regeneration proves to be an effective strategy in preventing traumatic neuroma.” I do not understrand this at all. The purpose of capping a nerve is to prevent the axons from growing out from the stump and forming a neuroma. How would capping a nerve, regardless of the nature of the cap, create a favorable microenvironment for regeneration?

Line 568, “It (neuroma) commonly occurs in patients with long-segment peripheral nerve injuries or amputations.” The concept of long-segment nerve injuries is mentioned several time in this paper. In practice, most neuromas occur across short segments of nerve. Whether the injury segment is long or short is probably irerelevant for neuroma formation, so the repetitive use of this term in the manuscript may just confuse the reader.

Round 2

Reviewer 2 Report

Comments and Suggestions for Authors

Line 25, “In conclusion, the utilization of tissue-engineered materials in treatment strategies significantly reduces the risk of traumatic neuroma post-peripheral nerve injury and introduces novel approaches for its treatment.” These materials are untested in humans, so this claim is premature. One could instead state, “In conclusion, the utilization of tissue-engineered materials has the potential to significantly reduce the risk of neuroma recurrence after surgical treatment.”

Line 51, “Advanced therapeutic techniques like transcutaneous electrical nerve stimulation (TENS) and laser therapy contribute to pain modulation and enhancement of neural function.” Laser therapy is probably a sham treatment. It would be better to mention evidence-based, effective therapies here instead, such as peripheral nerve stimulation, spinal cord stimulation, and/or dorsal root ganglion stimulation. 

Line 57, “Autograft and allograft techniques are now considered the “gold standard” for treating long-segment peripheral nerve injuries.” What is meant by “long-segment”? Usually these repair strategies are used for short segment injuries (<7 cm for allograft, <12 cm for autograft).

Line 105, axonotmetic injury does not result from lacerations. By definition, these injuries leave the structure of the nerve intact, except for the axons. Neurometic injuries occur when there is a laceration to the nerve. 

Line 114, this sentence starts out with the intent on describing three types of neuromas, but only one is mentioned in the sentence. I would instead rewcommend completing your points on the three types of injury first (neurapraxia, axonotmesis, neurotmesis). Then in the next paragraph, start your explanation of the Sunderland classification.

Line 128, here we go with the term “long segment” again. Is 5 mm really a “long segment”?

Line 245, use “Nonsurgical Treatments” as the subheading.

Line 251, delete “six”. There are actually many more than six.

Linbe 254, what is meant by “earliest”? Oldest? First-line treatment in any given patient?

Line 277, “scholars”? Not the best term here.

Line 281, resection and end-to-end anastomosis is no the only option here. It is rarely used in this circumstance, as the gap is usually too big. The more common used options that establish nerve continuity are autograft and allograft repair.

Line 287, be careful with the term “gold-standard” here. One could argue that the gold-standard for sensory nerve reconstructiuon is allograft repair as it avoids the complications you mention later in the paragraph.

Line 294, this value likely far underestimates the actual nbeuroma recurrence rate.

Line 306, the advantages of RPNI and TMR, for the purposes of this paper, are not about the control of prosthetic limbs. I would recommend focusing on the concept that these treatments redirect the regenerating axons into a permissive environment (denervated muscle, not enginneered scaffolds) that disperses the axons, thus preventing neuroma formation. Beginning a discussion of prosthetic limb control will totally confuse the reader.

Line 311, this should be the start of a new section and not lumped in with the prior section.

Line 331, what is meant by “suturing”? Please instead refer back to one or more actual treatment strategies you described previously (allograft, autograft, TMR, whatever).

Line 430, how does nerve capping guide axonal growth along desired patheways? Doesn’t it do the exact opposite?

Line 617, why exclude neuroma in continuity here? This is probably the most common type of neuroma.

Round 3

Reviewer 2 Report

Comments and Suggestions for Authors

Here are a few recommended, minor corrections:

Line 74, “Autograft and allograft techniques are currently regarded as the “gold standard” for treating peripheral nerve defects exceeding 3cm in length” should be changed to, “Autograft and allograft techniques are currently regarded as the “gold standard” for treating repairable peripheral nerve defects where a tension-free, end-to-end anastomosis is not feasible.”

Line 427, “Recently, some medical doctor have utilized ultrasound-guided local injections of anesthetic drugs and steroids” should be, “Local anesthetic and/or steroid injections along painful neuromas may provide pain relief, although the results are usually temporary.”

Line 454-467, in this paragraph, “RPN” should be “RPNI”.

Line 893, “It is classified as neuroma-in-continuity caused by transection injury and as amputation or terminal neuroma caused by amputation injury.” Delete this sentence, as it makes no sense and adds nothing to the paragraph.
